# Fatigue Life of 7005 Aluminum Alloy Cruciform Joint Considering Welding Residual Stress

**DOI:** 10.3390/ma14051253

**Published:** 2021-03-06

**Authors:** Jianxiao Ma, Ping Wang, Hongyuan Fang

**Affiliations:** 1State Key Laboratory of Nuclear Power Safety Monitoring Technology and Equipment, China Nuclear Power Engineering Co., Ltd., Shenzhen 518172, China; Jianxiaoma@outlook.com; 2State Key Laboratory of Advanced Welding and Joining, Harbin Institute of Technology, Harbin 150001, China; hyfang@hit.edu.cn; 3School of Ocean Engineering, Harbin Institute of Technology, Weihai 264209, China

**Keywords:** total fatigue life, cruciform join, welding residual stress, stress intensity factor, stress concentration

## Abstract

An evaluation method is proposed for determining the full fatigue life of aluminum alloy cruciform joint, including the crack initiation and propagation with welding residual stress. The results of simulations have shown that the boundary between the initiation and propagation stage is not constant, but a variable value. The residual stress leads to a significant reduction in both stages, which is more severe on initiation. With considering residual stress, the ratio of crack initiation to total life is below 7%. The effect of residual stress varies with external loading; when external load is lower, the residual stress has a greater effect.

## 1. Introduction

Cruciform joint can be used to connect two parts which are perpendicular to each other with fillet welds. Usually, the arc welding process is applied in this welding joint, and the axial force and bending moment can also be well transmitted. Therefore, it has important applications in bridges, high-speed trains, and ships [1]; these various types of mechanical structures tend to continue to serve under fatigue loads. However, due to the irregular geometric shape and the uneven surface, the cruciform joint is often one of the weak points. When a cruciform joint is not fully penetrated, there are often two vulnerable locations, namely welding toe and welding root [2,3]. It has been pointed out that it is possible to make the welding toe the weakest position of the entire cruciform joint by optimizing the weld size and process of the filler welding [4], that is to say, under any load conditions, the welding root will not be damaged before the welding toe if the weld is properly designed [5]. Thus, the fatigue failure of the welding toe needs more attention to obtain qualified fatigue life.

Until now, several semi-empirical methods based on the fatigue tests have been widely used in the industry. The examples include the widely accepted nominal stress [6,7,8], traction-based structural stress [9,10], hot spot stress [11,12,13,14], peak stress method [15], and similar assessment methods [16,17,18]. The above methods have considerable advantages, that is the accuracy of the experimental data which can be used in various fields and to formulate relevant industry standards [19,20]. However, those only consider the externally applied stress and are used to directly obtain the S-N curve from the test data, which often have theoretical shortcomings because of the neglect of welding residual stress.

With further research on welding fatigue, scholars have shifted from publishing these data to exploring the mechanisms behind them. Studies have shown that, two important segments should be accounted for when analyzing the fatigue life of welded structures [21,22]. The first is the stress concentration [23,24], and the second is the residual welding stress [25]. The superposition of the two factors makes the direct prediction of the welded structure fatigue life more difficult. In addition, for cruciform joint, whether there is a crack in the initial stage is also a question that is worth to be studied. Some studies support that this stage does not exist. Additionally, fracture mechanics is used directly to calculate fatigue life [25,26]. Moreover, even if the life obtained by the fracture mechanics calculation (such as Paris equation) is less than the test data, the calculation result can be adjusted to be consistent with the test by changing the lower limit of the cycle life integration, because in theory, when the lower limit of integration is zero, the life is infinite.

In order to solve the above problems, the impact of residual stress on the crack initiation and the propagation stages, and stress concentration should be included, and the boundary between these two stages must be quantified. The effect of residual stress has been extensively studied. In the studies by Ohta et al. [27], the transverse residual stress distributed along the weld was used to analyze this effect. The authors concluded that the effect of residual stress on fatigue propagation is mainly controlled by the macroscopic stress at the crack instead of using the stress intensity factor to describe the state of the crack. Lautrou et al. [28] and Dong et al. [29] dealt with the effects of residual stress and stress concentration on the crack initiation stage. In their papers, stress was considered to be the control parameter that affected the crack initiation, and the crack propagation stage was not included. Additionally, it should be noted that when the specimen is cut into the desired shape, the distribution of residual stress has a large deviation from the as-welded residual stress [30]. Residual stress has also been studied from the perspective of damage mechanics [31]. As the crack grows, the residual stress in the structure changes dynamically, so it is necessary to describe the residual stress field at different crack sizes. Terada et al. [32] acquired and analyzed such residual stress, and used the stress intensity factor as a parameter to characterize the residual stress from the perspective of fracture mechanics.

The theoretical models of the crack initiation and propagation stages are different, from the perspective of quantitative calculation of fatigue life, it is necessary to distinguish the boundary between the two stages which are the crack initiation and propagation stages. The Manson–Coffin or Morrow equation is generally used as a fundamental expression when determining the initial life. On the other hand, Paris law based empirical fitting formulas were developed for estimations of propagation life, such as the ones by Walker [33] and Forman [34], among others. However, since there is no apparent length limit in the crack initiation segment, the initial crack size in fatigue life calculations based on fracture mechanics is empirical [35,36]. As suggested by Mikulski [37], values of 0.1 mm to 0.5 mm are recommended. Fortunately, a method for the calculation of the boundary between the crack initiation and propagation is provided by Socie [38]. However, its application in welded structures with residual stress requires further considerations, which are presented in the paper.

In this paper, a method for quantitatively calculating the full fatigue life of the weld toe of the cruciform joint was established, which included the effects of residual stress and stress concentration, and the boundary size between the crack initiation and propagation stages was also analyzed instead of choosing a fixed constant. The full fatigue life of 7005 aluminum alloy cruciform joint is calculated as an application of the method proposed above. Welding finite element model and extended finite element method (XFEM) are used separately. The ratio of the initial life to the total life is analyzed with and without residual stress, which can provide a reasonable explanation for ignoring the life of crack initial stage. The calculation process provides a relatively comprehensive and novel idea for the calculation of total fatigue life of aluminum alloy cruciform joint considering welding residual stress, and it also explained the percentage of the initial life in the total life under the influence of residual stress, which can provide a theoretical support to ignore the crack initiation stage. The entire computing system also can be extended to any other welded joint.

## 2. Framework and Experiments

### 2.1. Framework Description

At the welding toe, the combination of stress concentration and residual welding stress increases the complexity of the calculation of fatigue life. Regarding the cruciform joint residual stress, it has different effects on the crack initiation and propagation. For the former, the crack initiation, the effect of residual stress is mostly noted in the stress concentration [39], While in the latter, the crack propagation, its effect is mostly found in the stress intensity factor [40,41]. The relationship between the various calculation steps is shown in Figure 1, it can be seen that the calculation process consists of three parts. Those parts include the calculation of residual stress (and related parameters), the calculation of the boundary between the crack initiation and propagation stage, and finally, the crack initiation and propagation life. The above-presented steps show that the residual stress and the boundary sizes of both stages are used as inputs for the fatigue life calculation. The boundary between crack initiation and propagation stages not only provides the calculated width/size for the crack initiation but is also a rational integral starting point for the crack propagation. The necessary calculation processes are presented in the following sections.

### 2.2. Formulas of Crack Initiation and Propagation Life

As proven in the majority of studies, the control mechanisms of crack initiation and the propagation stages are different [42,43,44]. Thus, calculation models are significantly different. Suppose that the boundary between crack initiation and propagation is when the crack size reaches *a*_0_. Coincidentally, the crack initiation stage is defined as the part before the crack size reaches *a*_0_.

The crack initiation stage could be regarded as a strain-controlled mechanism [45], allowing the use of the Manson–Coffin equation [46] to calculate the crack initiation life without the mean stress:(1)$$\frac{\Delta \varepsilon }{2}=\varepsilon_{f}^{\prime }{\left(2N_{f}\right)}^{c}+\frac{\sigma_{f}^{\prime }}{E}{\left(2N_{f}\right)}^{b}$$
where *Δ**ε* is the range of cyclic strain, *N_f_* is the fatigue life, $\sigma_{f}^{\prime }\;$ is the fatigue strength coefficient, *b* is the fatigue strength exponent, $\varepsilon_{f}^{\prime }$ is the fatigue ductility coefficient, *c* is the fatigue ductility exponent, and *E* is the Young modulus. In the context of cruciform joints with residual stress, the Manson–Coffin equation has two shortcomings. Firstly, the equation does not consider the effect of residual stress, and secondly, the expression cannot be directly applied due to the welding toe strain gradient. To mitigate the first issue, the Morrow equation can be used as an alternative:(2)$$\frac{\Delta \varepsilon }{2}=\varepsilon_{f}^{\prime }{\left(2N_{f}\right)}^{c}+\frac{\sigma_{f}^{\prime }-\sigma_{0}}{E}{\left(2N_{f}\right)}^{b}$$
where *σ*_0_ is the average stress, which allows the introduction of the residual stress.

Regarding the second issue, the integral method can be used to calculate the average initial life of the weld toe (for more details, see Figure 2). Due to stress concentration, *Δ**ε* is a function of the coordinate *x* along the direction of crack propagation, which can be written as *Δε**(x)*. Similarly, *N_f_* is also corresponding to *x* and can be written as *N_f_*_(_*_x_*_)_. Furthermore, to include the influence of residual stress, *σ*_0_ is expressed as *σ*_0_(*x*). Therefore, the crack initiation life *N_ini_* can be written as follows:(3)$$N_{ini}=\frac{\int_{0}^{a_{0}}N_{f}\left(x\right)dx}{a_{0}}$$

*N_f_* cannot be analytically solved as a function of *Δ**ε* by Equation (2), therefore, *N_f_*_(x)_ has no analytical solution either. The solving process depends on the presence of yield deformation at the welding toe. When the yield deformation is absent, the first segment to the right of the equal sign can be disregarded. Similarly, if there is yield deformation at the weld toe, the latter part of the equation is not considered. It is clear that if *Δ**ε*
*(x)* and *σ*_0_*(x)* are calculated, the *N_f_**_(x)_* can be solved via Equation (2). For this reason, calculating the *Δ**ε*
*(x)* and *σ*_0_*(x)* is critical when calculating the crack initiation life.

For a comparison, calculating the crack propagation life is a relatively easy task; the multi-parameter equations to calculate the crack growth rate are proposed in several studies. In this study, the Walker equation [46] is applied:(4)$$\frac{da}{dN}=C_{w}{\left[\frac{\Delta K}{{\left(1-R\right)}^{n}}\right]}^{m}$$
where *C*_*w*_, *m*, and *n* are material parameters, and *R* is the stress ratio. When there is residual stress in the structure, it is considered to be the effect of varying mean stress intensity factor [32]; according to the stress ratio definition [47], its influence can be included as follows:(5)$$R=\frac{K_{\min}}{K_{\max}}=\frac{K_{mF}+K_{e}-\frac{1}{2}\Delta K}{K_{mF}+K_{e}+\frac{1}{2}\Delta K}$$
where *K*_min_ and *K*_max_ are the minimum and maximum value of stress intensity factors at the crack tip, respectively, *K_mF_* is the stress intensity factor calculated for the average external load, Δ*K* is the stress intensity factor calculated for the range of external load, *K_e_* is the stress intensity factor calculated for the residual stress.

In order to find the crack propagation life *N_pro_*, the Equation (4) is integrated. After accounting for Equation (5), Equation (6) can be obtained as:(6)$$N_{pro}=\int_{a_{0}}^{t}\frac{1}{C_{w}}{\left(\frac{\Delta K{\left(2K_{mF}+2K_{e}+\Delta K\right)}^{n}}{{\left(2\Delta K\right)}^{n}}\right)}^{-m}da$$
where *t* is the plate thickness. Based on the presented equation, it can be concluded that, *K_mF_*, *K_e_*, and *ΔK* are vital when calculating the crack propagation life. Furthermore, it is convenient to notice that all three factors are functions of the coordinate *x.*

### 2.3. The Boundary between the Crack Initiation and Propagation

Socie presented a method for calculation of the boundary between the crack initiation and propagation [38], defined as *a*_0_ in Section 2.1. In the referenced method, the crack initiation and crack propagation rate curves are defined separately, while the boundary is determined by finding the intersection of the two curves.

The crack initiation rate curve could be expressed as $\frac{dx}{dN_{f}}$, which is also a function of *x*. However, Socie did not give the specific mathematical expression to calculate $\frac{dx}{dN_{f}}$ in the referenced paper. For this reason, the authors have to develop the expression in the following section.

The cruciform joint cross-section is shown in Figure 3; the high-stress concentration is found at the weld toe. The strain along the fatigue crack propagation direction can be written as *Δ**ε**(x)*, as shown in Figure 2. Additionally, its inverse function can be written as:(7)x=f(Δε)

The corresponding derivatives of Equation (7) with respect to *Δ**ε* are as follows:(8)$$\frac{dx}{d\Delta \varepsilon }=\frac{df\left(\Delta \varepsilon \right)}{d\Delta \varepsilon }=f^{’}\left(\Delta \varepsilon \right)$$

In addition to the previously described expressions, the Morrow equation is also drawn in Figure 3. It represents the functional relationship between *Δ**ε* and *N_f_*. Similarly, the *Δ**ε* derivative can be given as:(9)$$\frac{d\Delta \varepsilon }{dN_{f}}=2^{c+1}\varepsilon_{f}^{\prime }cN_{f}{}^{c-1}+2^{b+1}\frac{\sigma_{f}^{\prime }-\sigma_{0}}{E}bN_{f}{}^{b-1}$$

Lastly, the final result can be obtained by multiplying Equations (8) and (9):(10)$$\frac{dx}{dN_{f}}=\left[2^{c+1}\varepsilon_{f}^{\prime }cN_{f}{}^{c-1}+2^{b+1}\frac{\sigma_{f}^{\prime }-\sigma_{0}}{E}bN_{f}{}^{b-1}\right]\cdot f^{\prime }\left(\Delta \varepsilon \right)$$

$\frac{dx}{dN_{f}}$ represents the increase in crack length per cycle. Socie defined this parameter to be crack initiation rate.

Moreover, $\frac{da}{dN}$ in the Walker equation can also be expressed as a function of *x* due to *Δ**K*, *K_mF_*, and *K_e_*, all being functions of *x*; the concept of *x* is consistent with that of crack size *a*. Therefore, expressions ($\frac{da}{dN}$ − *x*) and ($\frac{dx}{dN_{f}}$ − *x*) can be drawn in the same coordinate system (shown in Figure 4). The two curves intersect at point. As Socie pointed out, at point A the rate of crack growth from the initiation mechanisms equals the rate of growth from the propagation mechanisms. When the boundary crack length $a_{0}$ is exceeded, the crack development will be described using the propagation processes. Finally, it must be noted that there is no analytical solution for finding point A abscissa. Thus, it must be solved via the numerical method.

### 2.4. Stress Concentration Model of the Crack Initiation and Propagation Stages

#### 2.4.1. Stress in the Vicinity of Weld Toe under the Tensile Load

The plane strain finite element model based on ABAQUS 6.13 have been established (shown in Figure 5) using the CPE8R finite elements. The cruciform joint plate thickness was set to 10 mm, and the welding fillet size is 13 mm, which is greater than the plate thickness to guarantee weld toe failure in fatigue assessment [2]. Grid density was increased and set to 0.01 mm at weld toe notch because the size of the crack initiation stage is very small.

In practice, the weld toe is often treated as a sharp notch corner, causing the mesh sensitivity [48]. The mesh sensitivity, in turn, caused the stress concentration factor (SCF) to be rather high. To avoid such problems, blunt notches at the weld toes were regarded as short arcs with various radii and based on the measurement of actual welding toes [16]. Four finite element models were established, with varying the toe arc radius; the values of 0.05 mm (grid size 0.005 mm), 0.1 mm, 0.2 mm, and 0.5 mm were used. The unit nominal stress was applied in all the models, which is the boundary condition of these static models. Lastly, the plane strain model thickness was 1 mm.

#### 2.4.2. Stress Intensity Factor of Crack Propagation Stage under the Tensile Load

Crack propagation rate can be described rather accurately by using the stress intensity factor. The cruciform joint can be simplified as a single-sided crack (see Figure 6). However, this simplification does not account for the local stress concentration at the weld toe. Therefore, the stress intensity factor of the weld toe under the tensile load can be expressed by Equation (11), allowing Equation (6) containing both the *K_mF_* and *Δ**K* to be solved efficiently. It should be noted that the crack size has been uniformly written as the coordinate *x.*
(11)$$K=\sigma S\sqrt{\pi x}M_{k}$$
where *σ* is the average cross-section stress and *S* is the shape factor equal to [49]:(12)$$S=\sqrt{\frac{2t}{\pi x}\tan\frac{\pi x}{2t}}\frac{0.752+2.02\left(\frac{x}{t}\right)+0.37{\left(1-\sin\frac{\pi x}{2t}\right)}^{3}}{\cos\frac{\pi x}{2t}}$$

*M_k_* is the parameter describing the weld toe stress concentration effect on the stress intensity factor (Equation (13)) [50].
(13)$$\begin{array}{c} M_{k}=C(\frac{a}{t})\\ C=0.8068-0.1554\left(\frac{H}{t}\right)+0.0429(\frac{H}{t})+0.0794\left(\frac{W}{t}\right)\\ k=-0.1993-0.1839\left(\frac{H}{t}\right)+0.0495(\frac{H}{t})+0.0815\left(\frac{W}{t}\right) \end{array}$$

### 2.5. Stress Intensity Factor Considering Welding Residual Stress

#### 2.5.1. Welding Finite Element Model of Cruciform Koint

The three-dimensional welding finite element model was developed to obtain the residual stress. As such, it allowed for the calculation of *σ*_0_ in Equation (10) and the stress intensity factor of residual stress *K_e_* in Equation (6).

The plate material applied in the finite element simulation was 7005 aluminum alloy, welded using ER 5356 filler wire. The chemical compositions are given in Table 1. The material parameters at 20 °C are shown in Table 2. The behavior of material properties as the temperature increases is nonlinear (shown in Figure 7).

The manual inert gas tungsten arc welding (GTAW) of the rolled aluminum alloy7005 was conducted, by using a weld filler wire of ER5356 type with a wire diameter of 1.2 mm. Four fillet welds are welded in turn in a clockwise direction, as show in Figure 8a, and the first and second weld pass have been marked; the plane of transverse stress is shown in Figure 8b) where the welding direction is also drawn. The welding process parameters are summarized in [51]. The heat source was set to be isothermal moving at the welding speed of 10 mm/s. Isothermal heat source means that the temperature of all positions of the filler metal is the same, and there is no temperature gradient, which is also the simplest welding heat source. Its temperature is 660 °C which is consistent with the melting point of the material. The distribution of transverse residual stress along the hypothetical crack propagation cross-section is presented in the following sub-section. For welding simulation, the element type is C3D8, and there are 26600 elements in total.

The residual stress is distributed throughout the welded structure and can be obtained by using the finite element method. Moreover, the extended finite element method (XFEM) has been used to calculate the stress intensity factor, which is generated by the residual stress field. Before the calculation, in order to ensure the validity of the XFEM model established in this paper, only external tensile loads were applied, and the results of XFEM and Equation (11) have been compared in Section 2.4.2.

The resulting finite element model is shown in Figure 9. The mesh is locally refined to 0.01 mm at the weld toe to ensure that minuscule cracks can be detected, allowing for the calculation of the stress intensity factor at various crack sizes. the element type is C3D8. The function of modeling cracks already exists in ABAQUS, which is achieved by inserting shell parts into solid parts. Number of contours for solving stress intensity factors is set to 10. A total of 14 models with different crack sizes were established (which are: 0.06 mm, 0.07 mm, 0.08 mm, 0.1 mm, 0.2 mm, 0.5 mm, 0.8 mm, 1 mm, 2 mm, 3 mm, 4 mm, 5 mm, 6 mm, 7 mm, 8 mm). The flow chart of calculating the residual stress intensity factor via XFEM is shown in Figure 10. The calculation of residual stress intensity factor requires the stress field of welding simulation. When there is a crack in the model, the residual stress field is applied first, mostly to ensure that the stress equilibrium state can be reached.

### 2.6. Fitting of the Manson–Coffin and the Walker Equations

As mentioned in Section 2.5.1, the material used in the simulation is 7005 aluminum alloy. The round bar specimens and compact tensile (CT) specimens were tested to obtain the necessary material parameters for life calculation. The specimen dimensions are shown in Figure 11a,c and the dimension of the CT specimen is designed according to ASTM E399 [52]. The fatigue tests were carried out using the MTS 809 experimental rig (see Figure 11b), and the test results are shown in Table 3, the principle of selecting the strain amplitude is to make the data points fall on the curved position of the fitting curve as much as possible, so that the fitting result can be more accurate. The function is fitted to the experimental results, and the parameters of the fitted function are shown both in Figure 12 and Table 4. If failure at weld root, that is the crack is propagating through the welding throat, we should consider the effect of welding filler-wire microstructure on the fatigue behavior, as Gaur has point out in [53].

## 3. Results and Discussion

### 3.1. The Effect of Stress Concentration on the Cruciform Joint

The effect of arc radius (shown in Figure 5) on the results is shown in Figure 13. It can be seen that, as the weld toe arc radius decreases, the stress concentration factor (SCF) at the notch position increases. Consequently, when the arc radius approaches zero, the SCF approaches infinity. Furthermore, when the distance from the notch surface is above 0.08 mm, the effect of the arc radius becomes negligible.

In order to quantitatively describe stress concentration, the curves shown in Figure 13 are fitted to the results via exponential function with three parameters. The resulting equation is written as follows:(14)$$x=\sqrt[{0.55}]{\frac{\ln\left(\frac{E\varepsilon }{F}-\beta_{3}\right)-\beta_{2}}{-\beta_{1}}}$$

Or Equation (15):(15)$$\varepsilon =\frac{F}{E}\exp\left(-\beta_{1}x^{0.55}+\beta_{2}\right)+\beta_{3}$$
where *F* is the tensile load, *ε* is the axial strain, and *β*_1_, *β*_2_, and *β*_3_ are functions of the toe notch radius *r* (please see Equation (16)).
(16)$$\begin{array}{c} \beta_{1}=-0{.8737\log}^{3}(r)-5{.2331\log}^{2}(r)-10.9497\log(r)-3.4779\\ \beta_{2}=-0{.1396\log}^{3}(r)-0{.8103\log}^{2}(r)-1.6371\log(r)-2.1951\\ \beta_{3}=-1{.9361r}_{0}{}^{3}+1{.2682r}_{0}{}^{2}-0{.2949r}_{0}+0.1294 \end{array}$$

### 3.2. Welding Residual Stress on Crack Plane and Its Stress Intensity Factor

The transverse stress distribution along the crack propagation plane is illustrated in Figure 14a,b. The former is the three-dimensional coordinate graph, while the latter is the projection graph. The location of the maximum stress gradient can be clearly seen, and its location is marked in Figure 14b. Furthermore, regarding the number of dimensions, it should be added that reducing the 3D to 2D space is a common simplification in fatigue analyses. Therefore, it is essential to find the distribution of the maximum transverse residual stress, along which the crack propagates downward from the initial position. The greatest stress gradient value is found at the position of 35 mm (the ordinate of Figure 14b). The relationship between the transverse residual stress and the displacement *x* is shown in Figure 14c), and the function is represented by a quartic polynomial, as shown in Equation (17):
(17)$$\sigma_{e}\left(x\right)=52012.8x^{4}-49059.4x^{3}+16517.34x^{2}-2459.1x+208.2$$
where *σ_e_(x)* is the transverse residual stress along the crack propagation plane. The mean stress from external load was not considered and the *σ_e_(x)* = *σ*_0_*(x)*.

Since the stress intensity factor of the residual stress field cannot be proved by experiment, the case of tensile external load is used to verify the validity of the established XFEM model. The comparison between the theoretical equation (Equation (11)) presented above and the stress intensity factor calculated via XFEM is shown in Figure 15. The abscissa is the crack size *a* while the ordinate is $\frac{K}{\sigma \sqrt{\pi a}}$. The results show that the presented analytical method is in agreement with the numerical simulation. Thus, it can be concluded that the established XFEM model is effective.

The effect of residual stress on the crack propagation can be best seen by observing the stress intensity factor. The XFEM calculation results of the model presented in Figure 9 are shown in Figure 16a, which also describes the parameter $K_{e}$ (Equation (6)). Based on the change of the stress intensity factor, the curve of residual stress intensity factor can be divided into three segments.

In segment I, the crack is small, and the stress intensity factor remains at a low level, and the release of residual stress is minimal. The very small crack size is the dominant factor at this segment. In segment II, as the cracks propagate, the stress intensity factor generated by the residual stress increases steeply, followed by a sharp decrease. The crack size corresponding to its maximum value is 2 mm. Furthermore, even though the lesser amount of the residual stress is released, the crack is more extensive than during segment I. The sufficient residual stress and crack size are the principal factors at this segment. Lastly, in segment III, the crack propagates further, and the residual stress is released more rapidly. Although the crack length is significant, there are no notable increases in the stress intensity factor. The significant release of residual stress is the dominant factor in this segment.

As the crack size increases, the contribution of residual stress gradually becomes lower. At the same time, the crack size contribution gradually becomes higher. The two trends are opposite, leading to a maximum value of the curve; the further explanation has been shown in Figure 16b. The stress intensity factor under unit residual stress field can be calculated, as it is only related to the distribution of the residual stress (unit residual stress field can be defined by dividing the maximum stress value to obtain a dimensionless stress field). It can be seen that as the crack grows, the residual stress is gradually released, and the maximum value of the residual stress gradually decreases. When the crack penetrates the entire plate thickness, the effect of the residual stress becomes zero. However, the stress intensity factor under the unit residual stress field gradually increases (as shown in Figure 16b). Finally, a more detailed fit was carried out to enable the application of the results to the subsequent calculations. The resulting expression is written as Equation (18):
(18)$$K_{e}=\exp(100.8\rho^{5}-256.6\rho^{4}+240.5\rho^{3}-97.8\rho^{2}+15.9\rho +3.83)$$
where $\rho =\cos^{10}[\frac{\pi }{12}(a-2)]$, which made Equation (18) more concise, *a* is the crack length, and *K*_e_ is the residual stress intensity factor.

### 3.3. The Boundary of Crack Initiation and Propagation for the Cruciform Joint

Assuming that the mean external load value is equal to zero and the weld toe arc radius is 0.05 mm, the parameters above are used to analyze the boundaries of crack initiation and propagation stages. The result has been shown in Figure 17a), it can be seen that $\frac{da}{dN}-x$ and $\frac{dx}{dN_{f}}-x$ intersect at specific points. Those points are the boundaries of crack initiation and propagation under different external loads. It can also be seen that the boundary size is not constant (if using Socie’s definition). Still, as the external load gradually increases, the boundary size follows. The boundary size is around 0.1 mm, which is similar to the results presented in the related studies [55,56].

The interpretation about the law of the change of the boundary size has been shown in Figure 17b. When the external load increases, both $\frac{da}{dN}-x$ and $\frac{dx}{dN_{f}}-x$ are translated along the positive direction of the y-axis. However, when the distance of $\frac{dx}{dN_{f}}$ translation is greater than the translation of $\frac{da}{dN}$, the intersection point is translated to the upper right corner. This means that the increase in stress has a greater impact on the crack initiation stage (when compared to the crack propagation), which is consistent with the traditional fatigue theory. Based on the analysis of related expressions, a simpler explanation can be provided. The functional relationship between the $\frac{da}{dN}/\frac{dx}{dN_{f}}$ and the external load *F* can be expressed through Equation (19), since $m-2+\frac{1}{b}<0$, so $\frac{da}{dN}/\frac{dx}{dN_{f}}<1$.
(19)$$\frac{\frac{da}{dN}}{\frac{dx}{dN_{f}}}=\frac{C_{w}{\left[\frac{\Delta K}{{\left(1-R\right)}^{n}}\right]}^{m}}{2^{b+1}\frac{\sigma_{f}^{\prime }-\sigma_{0}}{E}bN_{f}{}^{b-1}\cdot f^{\prime }\left(\Delta \varepsilon \right)}\propto \frac{F^{m}}{F^{\frac{b-1}{b}}\cdot F}\propto F^{m-2+\frac{1}{b}}<1$$
where the first term of Morrow equation is not considered, because the degree of $N_{f}$ in the first item is c−1, which is less than zero, and $N_{f}$ is relatively large, so $N_{f}{}^{c-1}$ can be ignored.

### 3.4. Crack Initiation and Propagation Life

The S−N curves were compared both with and without the residual stress to illustrate its effects on fatigue initiation, propagation, and total life. The four cases with toe notch radii of *r* = 0.05 mm, *r* = 0.1 mm, *r* = 0.2 mm, and *r* = 0.5 mm were studied; the average external load was assumed to be zero.

The effect of the residual stress on the S-N curve during the crack initiation stage was plotted in the semi-log coordinate system; The results in the absence of residual stress are shown in Figure 18a, while the case with the residual stress is shown in Figure 18b. When the toe notch radius is large (0.5 mm), the S−N curve is skewed to the right from the other curves. That indicates that the larger notch radius reduces the stress concentration and increases the crack initiation life. Once the stress amplitude *Δσ* reaches 140 MPa, the crack initiation life is approximately 10^4^ cycles without the residual stress and 6000 with residual stress. Thus, it can be concluded that residual stress reduces the crack initiation life by nearly 40%. At the stress amplitude *Δσ* = 40 MPa, the life without residual stress has reached 10^7^ cycles, while the life with residual stress was below 10^5^. Thus, the difference between the two was above two orders of magnitude. Hence, the authors concluded that lower the stress amplitude, the greater the residual stress effect. The effect of the residual stress can be greatly reduced by increasing the stress amplitude.

The effect of the residual stress on the S−N curve in the crack propagation stage is shown in Figure 19. When the stress amplitude *Δσ* is 140 MPa, the component life under the influence of residual stress is 8·10^4^ cycles. When there is no residual stress, the component lasts for 2·10^5^ cycles. The difference between the two is 2.5 times. When the stress amplitude *Δσ* = 40 MPa, the life with residual stress is 2·10^6^ cycles and without 2·10^7^ cycles; the difference between the two is ten times. This shows that the residual stress has a greater effect at the low-stress amplitudes, but it is much smaller when compared to the crack initiation stage. It can be said that the crack initiation stage is more severely affected by residual stress. It can also be seen that the S−N curves of the crack propagation stage basically coincides regardless of the weld toe arc radius, indicating that the weld toe radius does not affect the crack propagation life.

The effect of residual stress on total fatigue life is shown in Figure 20. When the stress amplitude is high, if there is no residual stress, the S−N curves basically coincide at different notch radii, indicating that the proportion of crack initiation life, in this case, is trivial. As the stress amplitude decreases, the crack initiation life gradually increases; when the stress amplitude is 40 MPa the proportion of the crack initiation life becomes significant (see Figure 20a). The life of the crack initiation stage is hardly shown when the residual stress is present; the S−N curves of different notch radii coincide at all the stress amplitudes (see Figure 20b).

The residual stress effects on the total life change as the stress amplitudes vary (see Figure 21a). At the relatively low-stress amplitudes, such as 40 MPa, the residual stress will reduce the total fatigue life up to seven times (calculated for a notch radius of 0.05 mm). However, at the high-stress amplitude, such as 140 MPa, the residual stress halves the total fatigue life. Based on that, it can be concluded that an increase in stress amplitude diminishes the effect of residual stress. Furthermore, based on the comparison of different notch radii, it was observed that the stress concentration deteriorates as the notch radius increases. At the same time, the effect of residual stress gradually decreases.

In the context of the total fatigue life, the ratio of crack initiation to propagation life reiterates the importance of the initiation stage. Said ratio changes regardless of the presence of residual stress (as shown in Figure 21b). In the absence of residual stress, and at the low load levels, the crack initiation stage is rather significant. As the load increases, its importance gradually decreases. However, when there is residual stress, the ratio of the crack initiation to propagation is below 7%, indicating that the residual stress diminishes the importance of the crack initiation stage. In other words, in the cruciform weld joint, the total life is predominantly controlled by the crack propagation stage.

## 4. Conclusions

In the present work, the full fatigue life of the cruciform joint at weld toe has been calculated. The welded toe is considered as a circular arc notch, in which the stress curve is fitted by using the results of the cruciform joint’s static finite element analysis. Stress intensity factor of the residual stress has been obtained by using XFEM. Furthermore, in order to conduct the later calculations of fatigue life, the expressions for calculating the stress concentration and residual stress caused by the external load have been provided. Finally, the crack initiation and propagation life have been presented. The following conclusions can be drawn:

The boundary between the initiation and propagation stage is not a constant, but a variable value, which will increase when the external load rises. This phenomenon indicates that the microcrack propagation rate during the crack initiation stage is more sensitive to the applied external load than that at the crack propagation.In the context of the cruciform joint welded toe, the functional relationship between the residual stress intensity factor calculated by residual stress and the crack length is quite complex, whose functional curve during the period of crack propagation can be divided into three segments. In segment I, because of the rather small crack in the welding toe, although there is a large residual stress in the cruciform joint, stress intensity factor is still at a low level. In segment II, the crack size further increases, and sufficient residual stress is retained in the structure, thereby leading to the maximum of stress intensity factor. Finally, in segment III, whose phenomenon is that the crack size increases continuously, despite an increase in the crack size, the stress intensity factor drops to a low level again because of the release of the residual stress.The residual stress caused by welding process of the cruciform joint reduces the fatigue life at the crack initiation and propagation stages, respectively. However, the residual stress has a more significant impact on the crack initiation. The crack initiation stage only accounts for less than 7% of the total fatigue life based on selected metal material and welded joint. From the perspective of engineering practicability, the life of the crack initiation stage can be ignored without unacceptable deviations.

## Figures and Tables

**Figure 1 materials-14-01253-f001:**
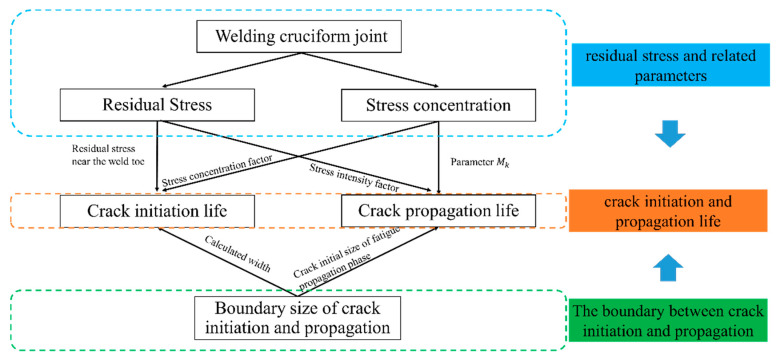
Flowchart of logical relationships between specific steps in order to calculate total fatigue life.

**Figure 2 materials-14-01253-f002:**
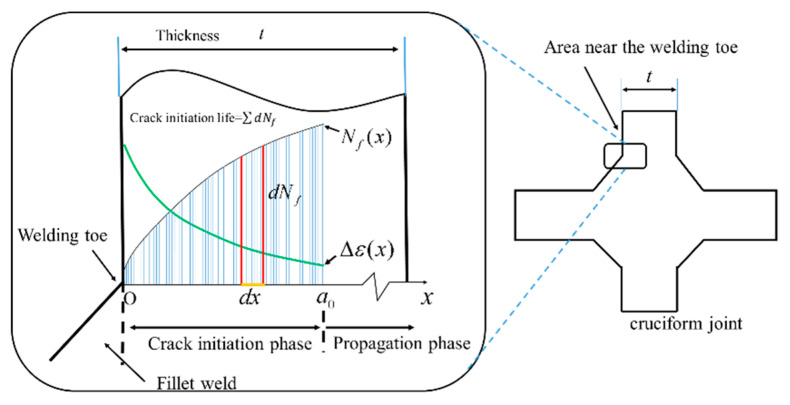
Calculation of weld toe crack initiation life of the cruciform joint.

**Figure 3 materials-14-01253-f003:**
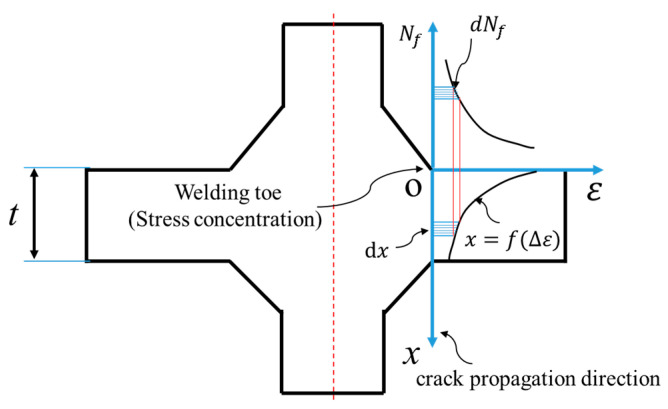
Cross section of cruciform joint with high stress concentration at the weld toe.

**Figure 4 materials-14-01253-f004:**
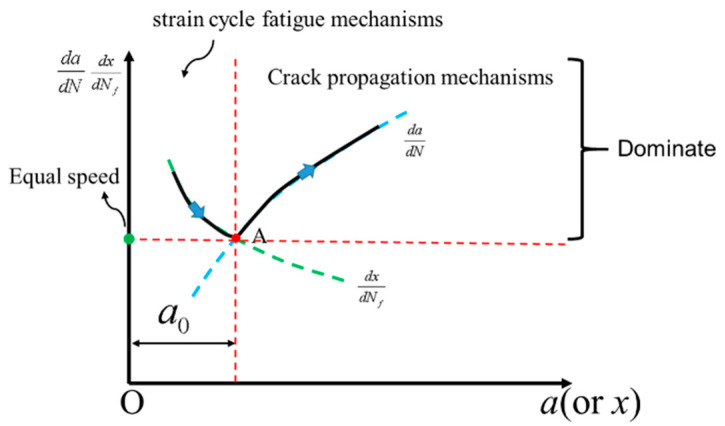
The boundary between the crack initiation and propagation determined by the initiation mechanisms and the propagation mechanisms.

**Figure 5 materials-14-01253-f005:**
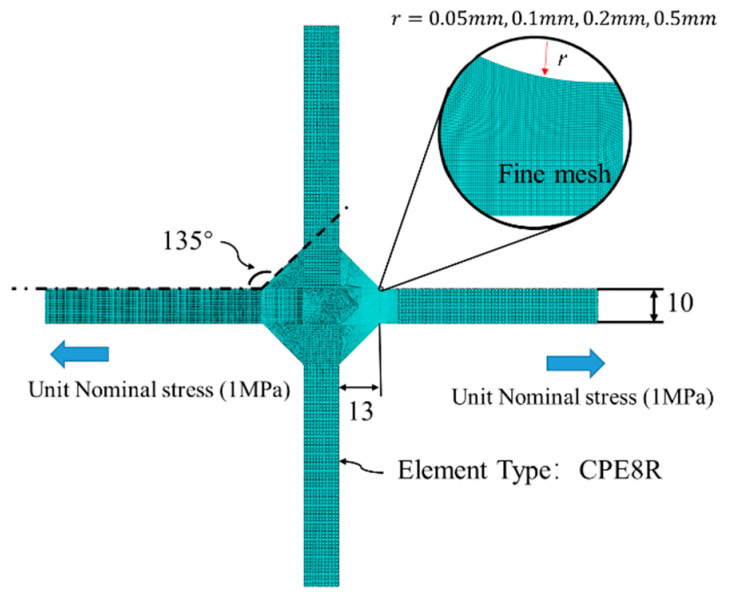
Plane strain model for static analysis, mesh refinement near the weld toe.

**Figure 6 materials-14-01253-f006:**
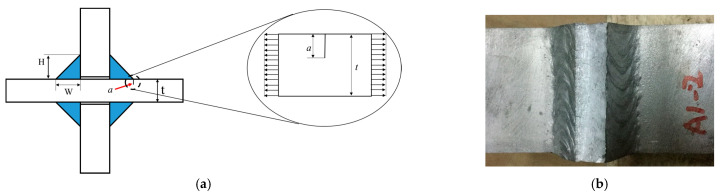
Weld toe cracks of the cruciform joint can be equivalent to a single-sided crack. (**a**) Illustration for crack at weld toe, (**b**) Fatigue fracture at weld toe.

**Figure 7 materials-14-01253-f007:**
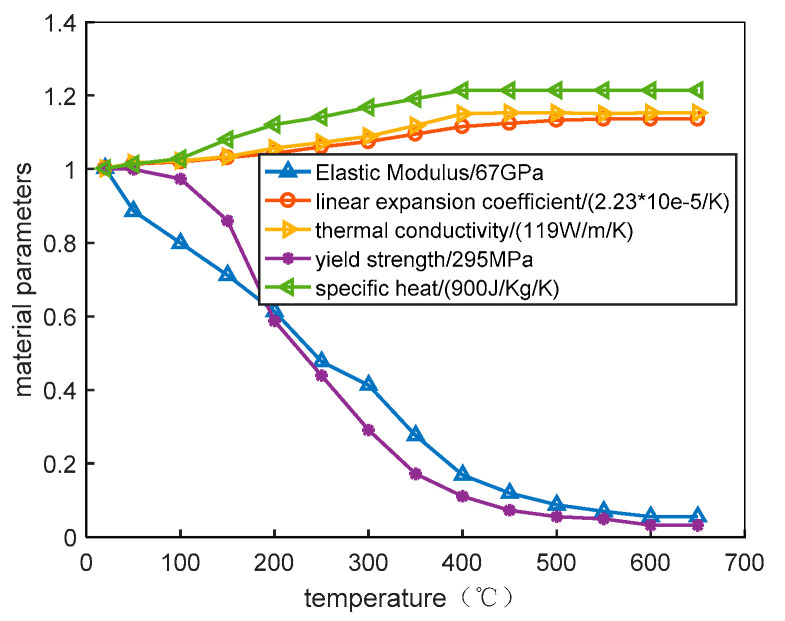
Material parameters of 7005 aluminum alloy as a function of temperature.

**Figure 8 materials-14-01253-f008:**
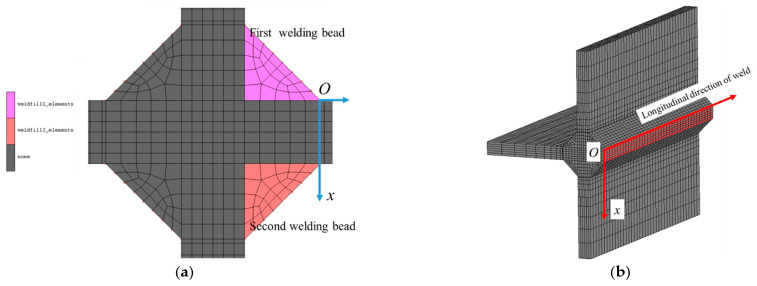
Weld bead and extraction path of transverse residual stress. (**a**) Schematic of bead fill for single pass welding; (**b**) Extraction path of transverse residual stress in crack propagation plane.

**Figure 9 materials-14-01253-f009:**
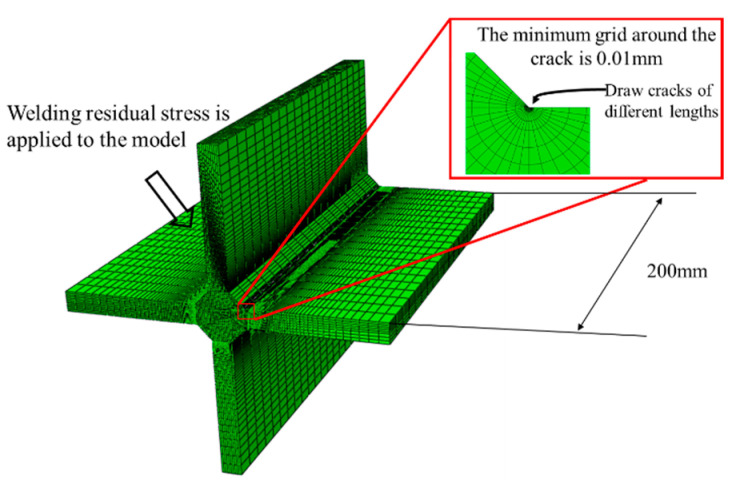
XFEM model of cross joint to obtain stress intensity factor with small crack.

**Figure 10 materials-14-01253-f010:**
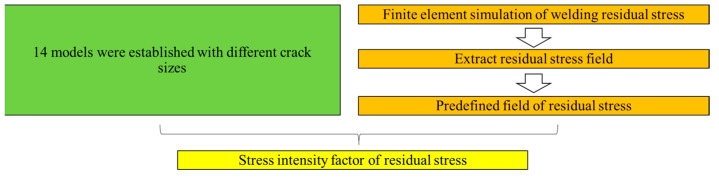
Flow chart for calculating the stress intensity factor of residual stress.

**Figure 11 materials-14-01253-f011:**
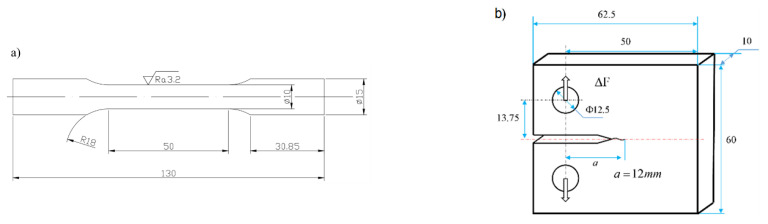
Processing and fatigue test of round bar and CT specimens (units, mm). (**a**) the dimensions of the round bar specimen. (**b**) the dimensions of the CT specimen [54].

**Figure 12 materials-14-01253-f012:**
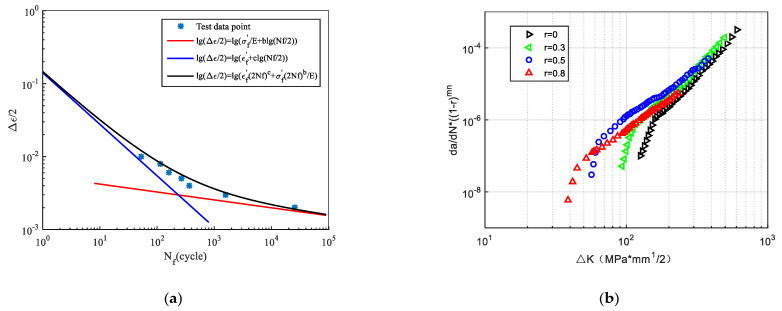
The fitting results of the Manson-Coffin and the Walker equations [54]. (**a**) The data fit for the Manson-Coffin equation; (**b**) The data fit well for the Walker equation.

**Figure 13 materials-14-01253-f013:**
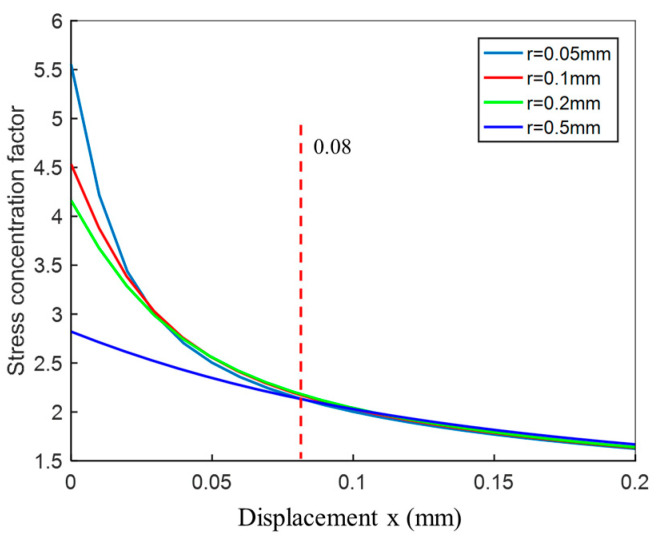
Stress with different toe notch radius under tensile load near weld toe.

**Figure 14 materials-14-01253-f014:**
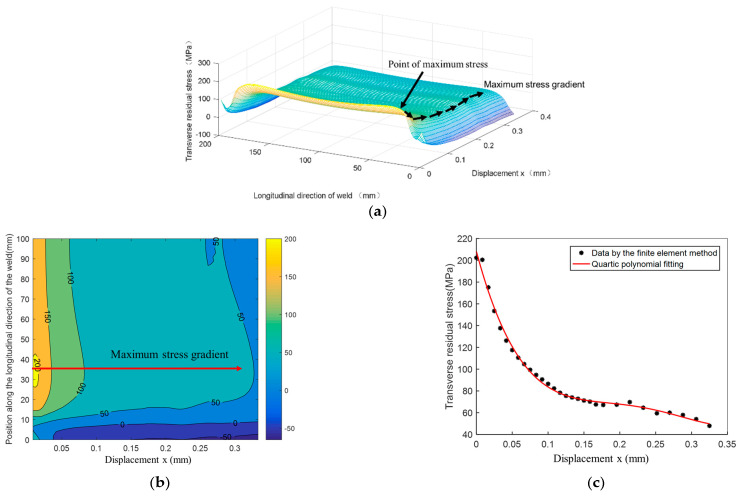
Results of welding residual stress distribution and fitting of maximum residual stress gradient. (**a**) Three-dimensional graph of residual stress on crack propagation surface; (**b**) Contour map of residual stress on crack propagation surface; (**c**) Fitting of transverse residual stress and displacement *x*.

**Figure 15 materials-14-01253-f015:**
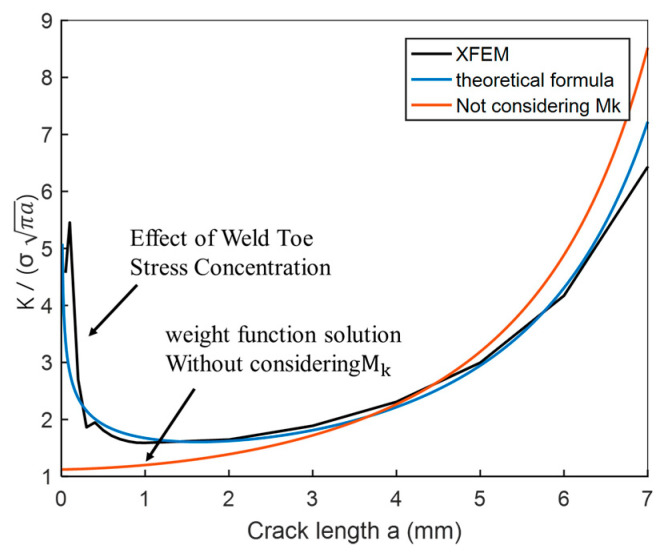
Comparison of calculation results of stress intensity factor with $M_{k}$ and XFEM method.

**Figure 16 materials-14-01253-f016:**
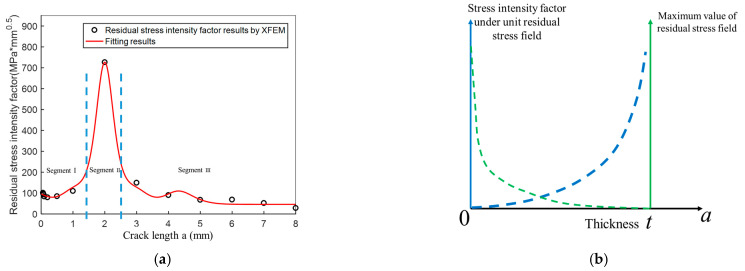
Functional relationship between residual stress intensity factor and crack length *a* as crack grows. (**a**) Residual stress intensity factor; (**b**) Variation of residual stress and stress intensity factor as crack grows.

**Figure 17 materials-14-01253-f017:**
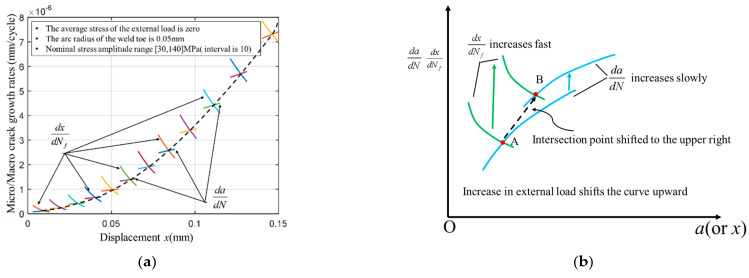
The boundary between crack initiation and propagation and its influencing factors. (**a**) The solution of the boundary and its relationship with nominal stress; (**b**) Influence of external load on $\frac{dx}{dN_{f}}$ and $\frac{da}{dN}$.

**Figure 18 materials-14-01253-f018:**
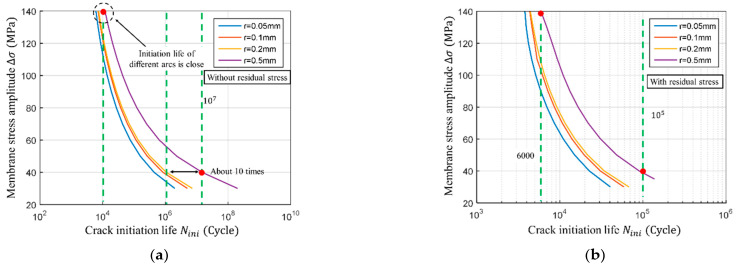
Results of S−N curve at crack initiation stage. (**a**) S−N curve of crack initiation with different notch radius without residual stress; (**b**) S−N curve of crack initiation with different notch radius with residual stress.

**Figure 19 materials-14-01253-f019:**
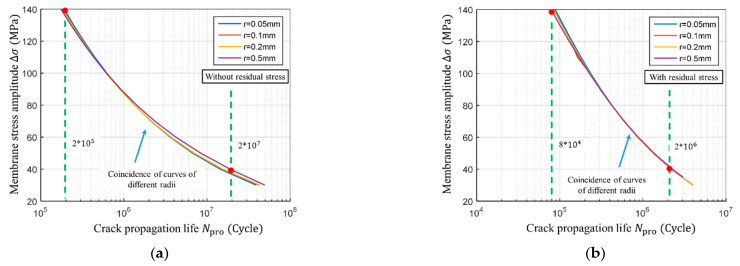
Calculation results of S−N curve at crack propagation stage. (**a**) S−N curve of crack propagation with different notch radius without residual stress; (**b**) S−N curve of crack propagation with different notch radius with residual stress.

**Figure 20 materials-14-01253-f020:**
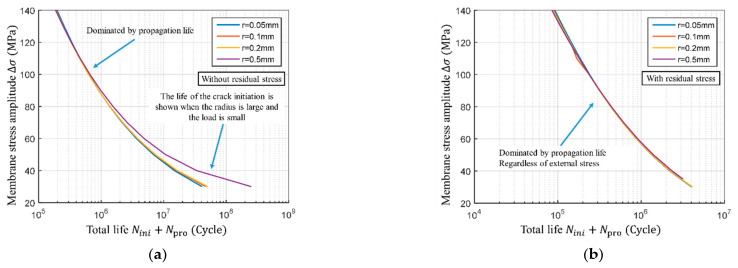
Calculation results of S−N curve of total fatigue life. (**a**) S−N curve of total life with different notch radius without residual stress; (**b**) S−N curve of total life with different notch radius with residual stress.

**Figure 21 materials-14-01253-f021:**
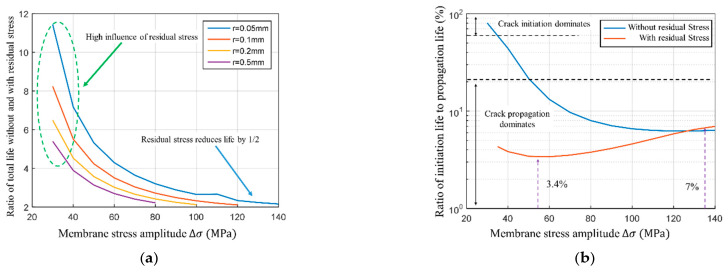
Effect of residual stress on total life and ratio of initiation life to propagation life. (**a**) Relationship between the degree of life reduction and membrane stress; (**b**) Relationship between ratio of initiation life to propagation life and membrane stress.

**Table 1 materials-14-01253-t001:** Chemical compositions of 7005 and ER5356 wt%.

Si	Fe	Cu	Mn	Mg	Zn	Ti	Cr	Al
0.07	0.15	0.15	0.35	1.2	4.4	0.05	0.18	Bal.
0.25	0.4	0.1	0.3	0.12	0.1	0.14	0.13	Bal.

**Table 2 materials-14-01253-t002:** Material property of base metal and filler at 20 °C.

Yield Strength/MPa	Thermal Conductivity/W·m^−1^·K^−1^	Specific Heat Capacity/mJ·t^−1^·K^−1^	Elastic Modulus/MPa	Thermal Expansion Coefficient/K^−1^	Density/t·m^−3^	Poisson’s Ratio
295	129	9.0 × 10^8^	6.7 × 10^4^	2.38×10^−5^	209	0.33
167	119	9.0 × 10^8^	6.7 × 10^4^	2.23×10^−5^	2.75 × 10^−9^	0.33

**Table 3 materials-14-01253-t003:** Strain fatigue test results [54].

Test Piece Number	Cyclic Strain Amplitude Δε/2(%)	$\mathrm{Cycles}\;N_{f}\;(\mathrm{Cycle})$
1#	0.2	25,427
2#	0.3	1563
3#	0.4	368
4#	0.5	264
5#	0.6	162
6#	0.8	113
7#	1	53

**Table 4 materials-14-01253-t004:** The value of the fitted parameters [54].

Parameters	$\varepsilon_{f}^{\prime }$	$\sigma_{f}^{\prime }$	b	c	$C_{w}$	n	m
fitted value	0.2298	388.6	−0.1083	−0.7063	3.766 × 10^−13^	0.561	3.111

## Data Availability

Not applicable.

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
