# Peer review of "Fatigue Life of 7005 Aluminum Alloy Cruciform Joint Considering Welding Residual Stress"

_materials, 2021, doi:10.3390/ma14051253_

Round 1
Reviewer 1 Report
The article is interesting, it can also be regarded as a continuation of the work "Quantitative analysis of overload effect at crack tip on crack growth rate of 7005 aluminum alloy", published in the International Journal of Fatigue. However, the work requires editing on some points:
(1) It is necessary to correct the heading numbers, as they are repeated.
(2) Many grammatical errors, for example (335) "arc radii" instead of "arc radius" etc.
(3) There are also annoying typos, for example, (461-462) «the life without (!) Residual stress has reached 10^7 cycles, while the life without (!) Residual stress was below 10^5»
(4) Formulas are very small and difficult to read.
(5) Conclusions should be rooted in specific clear points.
(6) Part of the data has already been published in the above article, for example, (324) - Fig. C1 in the International Journal of Fatigue, (328) - Table C1 in IJF, (332a) - Fig. C2 in IJF etc. Perhaps they should not be presented at all, but links should be made.
Author Response
First of all, the authors would like to use this opportunity to express their appreciation for the reviewers’ detailed comments. We found these comments are valuable and helpful for improving the quality of the paper, as well as clarifying significance of our research. We have studied the comments carefully and have revised our manuscript accordingly when appropriate. Below are the author’s item-by-item response including specific revisions made to the manuscript. For clarity, the reviewers’ comments are listed in italic, any new revisions incorporated in the revised manuscript are highlighted in yellow for Reviewer #1’s comments, highlight in blue for Reviewer #2’s comments, highlight in grey for Reviewer #3’s comments, and highlight in green for Reviewer #4’s comments. The article is interesting, it can also be regarded as a continuation of the work "Quantitative analysis of overload effect at crack tip on crack growth rate of 7005 aluminum alloy", published in the International Journal of Fatigue. However, the work requires editing on some points: (1) It is necessary to correct the heading numbers, as they are repeated. Our Response: The author thanks for pointing out the details, and the author has processed all the headings for the whole text. (2) Many grammatical errors, for example (335) "arc radii" instead of "arc radius" etc. Our Response: The author thanks for pointing out the details, and "arc radii" has been replaced with "arc radius", we also checked the whole article to avoid similar problems. (3) There are also annoying typos, for example, (461-462) «the life without (!) Residual stress has reached 10^7 cycles, while the life without (!) Residual stress was below 10^5» Our Response: The authors thanks for pointing out the details, the correct expression is: At the stress amplitude = 40 MPa, the life without residual stress has reached 107 cycles, while the life with residual stress was below 105. (4) Formulas are very small and difficult to read. Our Response: The authors thank for pointing out the details, we are very sorry for the inconvenience, the font in the formulas are enlarged to the same size as the text font. (5) Conclusions should be rooted in specific clear points. Our Response: The authors totally agree with the reviewer, and the conclusion has been rewritten as: In the present work, the full fatigue life of the cruciform joint at weld toe has been calculated. The welded toe is considered as a circular arc notch, in which the stress curve is fitted by using the results of the cruciform joint’s static finite element analysis. Stress intensity factor of the residual stress has been obtained by using XFEM. Furthermore, in order to conduct the later calculations of fatigue life, the expressions for calculating the stress concentration and residual stress caused by the external load have been provided. Finally, the crack initiation and propagation life have been presented. The following conclusions can be drawn: The boundary between the initiation and propagation stage is not a constant, but a variable value, which will increase when the external load rises. This phenomenon indicates that the microcrack propagation rate during the crack initiation stage is more sensitive to the applied external load than that at the crack propagation. In the context of the cruciform joint welded toe, the functional relationship between the residual stress intensity factor calculated by residual stress and the crack length is quite complex, whose functional curve during the period of crack propagation can be divided into three segments. In segment I, because of the rather small crack in welding toe, although there is a large residual stress in the cruciform joint, stress intensity factor is still at a low level. In segment II, the crack size further increases, and sufficient residual stress is retained in the structure, thereby leading to the maximum of stress intensity factor. Finally, in segment III, whose phenomenon is that the crack size increases continuously, despite a increase in the crack size, the stress intensity factor drops to a low level again because of the release of the residual stress. The residual stress caused by welding process of the cruciform joint reduces the fatigue life at the crack initiation and propagation stages respectively. However, the residual stress has a more significant impact on the crack initiation. The crack initiation stage only accounts for less than 7% of the total fatigue life based on selected metal material and welded joint. From the perspective of engineering practicability, the life of the crack initiation stage can be ignored without unacceptable deviations. (6) Part of the data has already been published in the above article, for example, (324) - Fig. C1 in the International Journal of Fatigue, (328) - Table C1 in IJF, (332a) - Fig. C2 in IJF etc. Perhaps they should not be presented at all, but links should be made. Our Response: The authors thank for pointing out the details. This study focus on the fatigue life of aluminum welded joint, so the Manson-Coffin curve and the Walker equation are very important( Fig 11c) CT specimen are used for fitting Walker equation ). Moreover, the author consider that it is more reasonable to show details by these pictures and table. Thus, to avoid inconveniences as well as duplicate publication, we have cited our article published in International Journal of Fatigue.

Reviewer 2 Report
General description
The manuscript concerns the problem of fatigue life calculation of welded cruciform joints. It is very well prepared paper supported by large and decent studies. The proposed model includes the most important factors influencing the fatigue life. Two very important issues (mostly neglected) were taken into account, i.e. residual stresses and two stages of fatigue damage development: crack initiation and crack propagation periods. The more, the study implements the finite element methods to identify the distribution of residual and applied stresses. The main disadvantage of the study is the lack of experimental tests on welded cruciform joints. Suggestions for paper improvement are given below:
Comments:
- Page 1, abstract. It is necessary to emphasis that the results of the study are obtained by simulation. Thus, in line 15 I proposed the following modification:
“The results of simulations have shown that the boundary between the initiation and propagation stage is not constant, but a variable value”.
- Page 1, line 32, there is “Fortunately, it has been pointed out that …” Why fortunately? I do not see the connection with further text. Especially with ending sentence: “Therefore, the fatigue failure of the welding toe needs more attention.” ?
- Page 1, lines 38-41, consider alternative methods as probabilistic models based on the weakest link approach presented e.g. in [Blacha L, Karolczuk A, Bański R, Stasiuk P. Application of the weakest link analysis to the area of fatigue design of steel welded joints. Eng Fail Anal 2013;35:665–77. doi:10.1016/j.engfailanal.2013.06.012.]
- Page 2, line 51, “…is the dynamic residual welding stress.” Why dynamic? I propose to remove “dynamic”
- Page 2, line 60, “… above problems as much as possible, the impact…”. Please delete this unnecessary phrase.
- Page 2, line 73, “Of course, for theoretical analysis, Residual … ”. Please delete this unnecessary phrase.
- Page 2, line 75, “the residual stress in the structure changes dynamically, so” Again, why dynamically? The residual stress is not just relaxing?
- Page 3, line 116, “The logical relationship between the various …”. Please delete this unnecessary phrase.
- Font size for equations is too small. Also, the numbering style of the equation is varying.
- Page 7, line 259, “… is the shape factor equal…” Reference to the applied form of shape factor is required.
- The model is fitted by experimental data based on uniaxial fatigue tests conducted on the raw material. The effect of microstructure changes due to the welding process is even considered (structural notch effect). Some comments and references for this problem are necessary. For example : [Gaur, V., Enoki, M., Okada, T., Yomogida, S., 2018. A study on fatigue behavior of MIG-welded Al-Mg alloy with different filler-wire materials under mean stress. International Journal of Fatigue 107, 119–129.. doi:10.1016/j.ijfatigue.2017.11.001]
Author Response
First of all, the authors would like to use this opportunity to express their appreciation for the reviewers’ detailed comments. We found these comments are valuable and helpful for improving the quality of the paper, as well as clarifying significance of our research. We have studied the comments carefully and have revised our manuscript accordingly when appropriate. Below are the author’s item-by-item response including specific revisions made to the manuscript. For clarity, the reviewers’ comments are listed in italic, any new revisions incorporated in the revised manuscript are highlighted in yellow for Reviewer #1’s comments, highlight in blue for Reviewer #2’s comments, highlight in grey for Reviewer #3’s comments, and highlight in green for Reviewer #4’s comments.
The manuscript concerns the problem of fatigue life calculation of welded cruciform joints. It is very well prepared paper supported by large and decent studies. The proposed model includes the most important factors influencing the fatigue life. Two very important issues (mostly neglected) were taken into account, i.e. residual stresses and two stages of fatigue damage development: crack initiation and crack propagation periods. The more, the study implements the finite element methods to identify the distribution of residual and applied stresses. The main disadvantage of the study is the lack of experimental tests on welded cruciform joints. Suggestions for paper improvement are given below:
Comments:
- Page 1, abstract. It is necessary to emphasis that the results of the study are obtained by simulation. Thus, in line 15 I proposed the following modification:
“The results of simulations have shown that the boundary between the initiation and propagation stage is not constant, but a variable value”.
Our Response: The authors thank for pointing out the details, and in the abstract, we has already made the revision.
- Page 1, line 32, there is “Fortunately, it has been pointed out that …” Why fortunately? I do not see the connection with further text. Especially with ending sentence: “Therefore, the fatigue failure of the welding toe needs more attention.” ?
Our Response: The authors thank for pointing out the details, for the word “fortunately”, we did not use it properly here, and it was deleted from the text for we could not come out with another expression.
For the last sentence, our origin expression was : “ With proper design, the cruciform joint would failure at the weld toe instead of weld root, thus more attention should be paid on the fatigue life of weld toe”, to make it clear, we have rewritten the sentence as: “Thus, the fatigue failure of the welding toe needs more attention to obtain qualified fatigue life.”
- Page 1, lines 38-41, consider alternative methods as probabilistic models based on the weakest link approach presented e.g. in [Blacha L, Karolczuk A, Bański R, Stasiuk P. Application of the weakest link analysis to the area of fatigue design of steel welded joints. Eng Fail Anal 2013;35:665–77. doi:10.1016/j.engfailanal.2013.06.012.]
Our Response: The authors thank for the reviewer’s advice on the probabilistic models, which provided a new perspective for us. We have referred in the article.
- Page 2, line 51, “…is the dynamic residual welding stress.” Why dynamic? I propose to remove “dynamic”
Our Response: The authors thank for pointing out the details, we removed the word“dynamic”.
- Page 2, line 60, “… above problems as much as possible, the impact…”. Please delete this unnecessary phrase.
Our Response: The authors thank for pointing out the details, we removed the unnecessary phrase.
- Page 2, line 73, “Of course, for theoretical analysis, Residual … ”. Please delete this unnecessary phrase.
Our Response: The authors thank for pointing out the details, we removed the unnecessary phrase.
- Page 2, line 75, “the residual stress in the structure changes dynamically, so” Again, why dynamically? The residual stress is not just relaxing?
Our Response: The authors thank for pointing out the details, here we tried to express the uneven distribution of the as-welded residual stress. And we have removed the words from the text.
- Page 3, line 116, “The logical relationship between the various …”. Please delete this unnecessary phrase.
Our Response: The authors thank for pointing out the details, we removed the unnecessary phrase.
- Font size for equations is too small. Also, the numbering style of the equation is varying.
Our Response: The authors thank for pointing out the details, as the 1# reviewer has pointed out this issue, we has processed already and marked the numbering in Yellow.
- Page 7, line 259, “… is the shape factor equal…” Reference to the applied form of shape factor is required.
Our Response: The authors thank for pointing out the details, Reference[49] has been added.
- The model is fitted by experimental data based on uniaxial fatigue tests conducted on the raw material. The effect of microstructure changes due to the welding process is even considered (structural notch effect). Some comments and references for this problem are necessary. For example : [Gaur, V., Enoki, M., Okada, T., Yomogida, S., 2018. A study on fatigue behavior of MIG-welded Al-Mg alloy with different filler-wire materials under mean stress. International Journal of Fatigue 107, 119–129.. doi:10.1016/j.ijfatigue.2017.11.001]
Our Response: The authors totally agree with the reviewer, different filler-wire materials has its effect on fatigue behaviors with round bar specimens. For the failure in our study, when the crack starts at the weld toe, it would propagate through the plate. On the cracking path, all of the material are base metal. It might be inaccurate to a certain extent, to make it clear, we might try to do tests with weld later. And we have added the article as ref[53]. {Page 10, last paragraph}

Reviewer 3 Report
- No explanation of symbols and abbreviations as E and others.
- It would be good to give the chemical composition of the material tested.
- It is a pity that the authors did not show photos of cracks or fracture, which would enrich the work.
- In Eqs. (11) and (12) should the factors be under a root, such as Mk?
- Please expand the information on the FEM calculations. What material model was used, number of finite elements, etc.?
- Fig. 11 - in the figure caption, please indicate that the dimensions are in mm on all drawings.
- It would also be worthwhile to quote the following papers: 1) Rozumek D., Lewandowski J., Lesiuk G., Correia J., The influence of heat treatment on the behavior of fatigue crack growth in welded joints made of S355 under bending loading. Int. J. of Fatigue 131, 2020, 2) ASTM E 739-80, Standard practice for statistical analysis of linearized stress-life (S-N) and strain-life (E-N) fatigue data, in: Annual Book of ASTM Standards, Vol.03.01. Philadelphia, 1989, pp. 667-673.
Author Response
First of all, the authors would like to use this opportunity to express their appreciation for the reviewers’ detailed comments. We found these comments are valuable and helpful for improving the quality of the paper, as well as clarifying significance of our research. We have studied the comments carefully and have revised our manuscript accordingly when appropriate. Below are the author’s item-by-item response including specific revisions made to the manuscript. For clarity, the reviewers’ comments are listed in italic, any new revisions incorporated in the revised manuscript are highlighted in yellow for Reviewer #1’s comments, highlight in blue for Reviewer #2’s comments, highlight in grey for Reviewer #3’s comments, and highlight in green for Reviewer #4’s comments.
- 1 No explanation of symbols and abbreviations as E and others.
Our Response: The authors thank for pointing out the details, in the study E is the Young modulus, please check line 141.
- It would be good to give the chemical composition of the material tested.
Our Response: The authors thank for pointing out the details, the chemical compositions have been added as Table 1.
- It is a pity that the authors did not show photos of cracks or fracture, which would enrich the work.
Our Response: The authors thank for pointing out the details, the picture has been added as Figure 6b).
- In Eqs. (11) and (12) should the factors be under a root, such as Mk?
Our Response: The authors thank for pointing out the details, here we treat the weld toe crack as a single-sided crack, and Eqs(11) and (12) are parameters for calculating Mk.
- Please expand the information on the FEM calculations. What material model was used, number of finite elements, etc.?
Our Response: The authors thank for pointing out the details, the base material in the model is 7005 aluminum alloy, and filler wire is ER 5356. And For welding simulation, the element type are C3D8, and there are 26600 elements in total. Also other relevant content has been added, please check Page 9 and Page10.
- Fig. 11 - in the figure caption, please indicate that the dimensions are in mm on all drawings.
Our Response: The authors thank for pointing out the details, we added the “unit,mm” in the caption of Figure 11
- It would also be worthwhile to quote the following papers: 1) Rozumek D., Lewandowski J., Lesiuk G., Correia J., The influence of heat treatment on the behavior of fatigue crack growth in welded joints made of S355 under bending loading. Int. J. of Fatigue 131, 2020, 2) ASTM E 739-80, Standard practice for statistical analysis of linearized stress-life (S-N) and strain-life (E-N) fatigue data, in: Annual Book of ASTM Standards, Vol.03.01. Philadelphia, 1989, pp. 667-673.
Our Response: The authors thank for recommending those papers, and we appreciate the reviewer’s advice and referred them in the text,check ref[23] and ref[24].

Reviewer 4 Report
1) Unchangeable mechanical properties were assumed. This would more or less apply for a soft annealed alloy. However, the yield strength given in Table 1 indicates that the alloy was precipitation hardened (condition T6) and as such very sensitive to overheating. Consequently, it should have been taken into account that elevated temperatures and especially overheating in the coarse grain heat affected zone detrimentally influence the mechanical properties of the base metal.
This matter should be addressed in the text.
2) Thermal conditions:
Only the welding speed was specified (line 279).
The consequences of this assumption (the influences on the results) should be should be discussed in the text.
Namely, the other crucial factor for the thermal conditions during welding is the heat input per millimeter. Both together, welding speed and heat input, influence the temperature fields, which are strongly time dependent.
3) Isothermal heat source (lines 278-281):
“Isothermal heat source means that the temperature of all positions of the filler metal is the same, and there is no temperature gradient, which is also the simplest welding heat source.”
Comment:
Does this mean that the temperature in each moment was assumed to be equal in the whole weld bead?
A discussion of influence of this simplification is needed.
Namely, in a real weld something like that is absolutely impossible. Consequently, the residual stress along the weld line is not almost constant and in the perpendicular direction it does not slowly decrease, as shown in Fig. 14 a. In reality, the residual stresses vary greatly in all directions - along the welding line and in a vertical direction. It even varies in the direction of the thickness of the base metal.And not only does the stress level vary, but also the compressive and tensile stresses alternate in all directions.
4)
“It´s temperature is 660℃ which is consistent with the melting point of the material.”
Comment
It is not quite so:
- 660 °C corresponds almost precisely to the melting point of pure aluminum.
- Alloys do not have a certain melting temperature but have a melting range.
- The alloy 7005 exhibits T solidus = 604 °C and T liquidus = 643 °C, while the T solidus and T liquidus of the filler material ER5356 are 571 °C and 635 °C, respectively (sources: http://www.matweb.com and https://www.harrisproductsgroup.com/~/media/Files/PDF/Spec-Sheets/Welding/Aluminum/5356Spec.pdf).
Consequently, residual stresses in a weld joint do not start to build up at 660 °C but at much lower temperatures. The consequences of this fact (the influences on the results) should be should be discussed in the text.
5) Lines 461-462:
“At the stress amplitude Δσ = 40 MPa, the life without residual stress has
reached 107 cycles, while the life without residual stress was below 105.”
Comment:
Something is wrong here: without – without.
6)
Lines 472-476:
“When the stress amplitude Δσ is 140 MPa, the component life under the influence of residual stress is 2·105 cycles. When there is no residual stress, the component lasts for 8·104 cycles. The difference between the two is 2.5 times. When the stress amplitude Δσ = 40 MPa, the life with residual stress is 2·107 cycles and without 2·106 cycles; the difference between the two is ten times.
Comment:
Is this true? How can it be that fatigue life is longer with tensile residual stress and shorter without residual stress? It seems illogical. Please explain!
7) In the text, the symbols of physical quantities are in general too small. The same applies for equations.
8) Figure 11 b is not essential. It could be omitted. However, if you choose to keep it in the manuscript, it should be much larger.
Author Response
First of all, the authors would like to use this opportunity to express their appreciation for the reviewers’ detailed comments. We found these comments are valuable and helpful for improving the quality of the paper, as well as clarifying significance of our research. We have studied the comments carefully and have revised our manuscript accordingly when appropriate. Below are the author’s item-by-item response including specific revisions made to the manuscript. For clarity, the reviewers’ comments are listed in italic, any new revisions incorporated in the revised manuscript are highlighted in yellow for Reviewer #1’s comments, highlight in blue for Reviewer #2’s comments, highlight in grey for Reviewer #3’s comments, and highlight in green for Reviewer #4’s comments.
- 1) Unchangeable mechanical properties were assumed. This would more or less apply for a soft annealed alloy. However, the yield strength given in Table 1 indicates that the alloy was precipitation hardened (condition T6) and as such very sensitive to overheating. Consequently, it should have been taken into account that elevated temperatures and especially overheating in the coarse grain heat affected zone detrimentally influence the mechanical properties of the base metal.
This matter should be addressed in the text.
Our Response: The authors thank for the reviewer’s advice on mechanical properties, we totally agree with the soft annealed phenomena in the coarse grain heat affected zone of aluminum alloy. In this study, we focus on the effect of residual stress on the fatigue at weld toe. The stress concentration at weld toe contributes to the crack initiation at the spot. We appreciate the reviewer’s advice, in the further work, we would emphasis on the soft annealed phenomena in the coarse grain heat affected zone in welding simulation of butt weld.
- 2) Thermal conditions:
Only the welding speed was specified (line 279).
The consequences of this assumption (the influences on the results) should be discussed in the text.
Namely, the other crucial factor for the thermal conditions during welding is the heat input per millimeter. Both together, welding speed and heat input, influence the temperature fields, which are strongly time dependent.
Our Response: The author thanks for pointing out the details, and relevant content has been added:
[Page 9 Paragraph 1] The manual inert gas tungsten arc welding (GTAW) of the rolled aluminum alloy 7005 was conducted using a weld filler wire of ER5356 type with a wire diameter of 1.2 mm. Four fillet welds are welded in turn in a clockwise direction, as show in Fig8 a, and the first and second weld pass have been marked out; the crack plane for transverse stress is shown in Fig. 8 b) where the welding direction is also drawn. The welding process parameters are summarized in ref[51].
- 3) Isothermal heat source (lines 278-281):
“Isothermal heat source means that the temperature of all positions of the filler metal is the same, and there is no temperature gradient, which is also the simplest welding heat source.”
Comment: Does this mean that the temperature in each moment was assumed to be equal in the whole weld bead?
A discussion of influence of this simplification is needed.
Namely, in a real weld something like that is absolutely impossible. Consequently, the residual stress along the weld line is not almost constant and in the perpendicular direction it does not slowly decrease, as shown in Fig. 14 a. In reality, the residual stresses vary greatly in all directions - along the welding line and in a vertical direction. It even varies in the direction of the thickness of the base metal. And not only does the stress level vary, but also the compressive and tensile stresses alternate in all directions.
Our Response: The author totally agree with the reviewer, It’s true that the temperature of all positions of the filler metal is not the same, thus we have the different heat resources models, such as the disc shaped weld heat source, double ellipsoidal weld heat source, et al. The calculated welding residual stress was used as an input for the XFEM analysis, thus we use a fixed melting temperature to make the process simpler and quicker, and the melting temperature could be set directly in Abaqus.
For the distribution of welding residual stress, here we will explain the Figure 14 in details. The longitudinal direction is the welding arc moving direction, the welding toe line in this direction are the potential cracking initiation spots. Figure 14 a) represents the three-dimensional transverse stress distribution along the crack propagation plane, the high stress starts from the surface. Figure 14b) is its projected version, where the transverse residual stresses vary greatly from the tensile status on the surface to the compressive stress 0.4 mm away. Then the transverse residual stress along the crack propagation plane is expressed as a quartic polynomial function σe(x) to calculate stress intensity factor K.
- 4) “It´s temperature is 660℃ which is consistent with the melting point of the material.”
Comment:It is not quite so: 660 °C corresponds almost precisely to the melting point of pure aluminum. Alloys do not have a certain melting temperature but have a melting range.
The alloy 7005 exhibits T solidus = 604 °C and T liquidus = 643 °C, while the T solidus and T liquidus of the filler material ER5356 are 571 °C and 635 °C, respectively (sources: http://www.matweb.com and https://www.harrisproductsgroup.com/~/media/Files/PDF/Spec-Sheets/Welding/Aluminum/5356 Spec.pdf).
Consequently, residual stresses in a weld joint do not start to build up at 660 °C but at much lower temperatures. The consequences of this fact (the influences on the results) should be discussed in the text.
Our Response: The author thanks for pointing out the details, we are totally agree with the reviewer’s opinion, the base metal 7005 and ER 5356 has their own melting points, and alloys do not have a certain melting temperature but have a melting range due to alloy compositions fluctuation.
In our study, the melting point is calculated based on the heat input from the welding arc, we use the following equation to get a theoretical melting point Tm=660 ℃:
Where c is the Specific heat capacity of filler material, m is weight of each bead(unit,kg), Tm is theoretical melting point, T0 is room temperature 20℃, Qi is linear heat input for each pass, li is welding length of each welding pass. As , where U is arc voltage, I is welding current, η is welding efficiency, here η=0.75 . Author has realized that the use of ‘melting point’ is not correct, so the sentence has been converted as: ‘Its temperature is 660℃ which can make the filler metal and part of the base material reach a molten state.’
The calculated welding residual stress was used as an input for the XFEM analysis, thus we use a fixed melting temperature to make the process simpler.
- 5) Lines 461-462: “At the stress amplitude Δσ = 40 MPa, the life without residual stress has reached 107 cycles, while the life without residual stress was below 105.”
Comment: Something is wrong here: without – without.
Our Response: The authors thanks for pointing out the typo, the correct expression is: At the stress amplitude = 40 MPa, the life without residual stress has reached 107 cycles, while the life with residual stress was below 105.
- 6) Lines 472-476: “When the stress amplitude Δσ is 140 MPa, the component life under the influence of residual stress is 2·105 cycles. When there is no residual stress, the component lasts for 8·104 cycles. The difference between the two is 2.5 times. When the stress amplitude Δσ = 40 MPa, the life with residual stress is 2·107 cycles and without 2·106 cycles; the difference between the two is ten times.
Comment: Is this true? How can it be that fatigue life is longer with tensile residual stress and shorter without residual stress? It seems illogical. Please explain!
Our Response: The authors thanks for pointing out the typos, the correct expression is: When the stress amplitude Δσ is 140 MPa, the component life under the influence of residual stress is 8·104 cycles. When there is no residual stress, the component lasts for 2·105 cycles. The difference between the two is 2.5 times. When the stress amplitude Δσ = 40 MPa, the life with residual stress is 2·106 cycles and without2·107 cycles; the difference between the two is ten times.
- 7) In the text, the symbols of physical quantities are in general too small. The same applies for equations.
Our Response: The authors thanks for pointing out the details, as other reviewer pointed out the same question, we have already changed the font size.
- 8) Figure 11 b is not essential. It could be omitted. However, if you choose to keep it in the manuscript, it should be much larger.
Our Response: The authors totally agree with the reviewer, and the Figure 11 b has been removed from the text.

Round 2
Reviewer 1 Report
Thank you very much for the correction. The work looks much better. In my opinion, there are only minor stylistic and grammatical edits, such as (393) - an extra point at the beginning of a sentence, etc
Author Response
The work looks much better. In my opinion, there are only minor stylistic and grammatical edits, such as (393) - an extra point at the beginning of a sentence, etc
Our Response: The author thanks for pointing out the details, and the author has removed the point and checked the whole text for the same typos.

Reviewer 3 Report
The authors took into account all comments and suggestions of the reviewer. I recommend publishing the article in the Materials.
In the description of table 1 it should be "chemical composition"
Author Response
In the description of table 1 it should be "chemical composition"
Our Response: The authors thank for pointing out the typos, and we have replaced with the "chemical composition" in Table 1.
